# The Influence of Air Nanobubbles on Controlling the Synthesis of Calcium Carbonate Crystals

**DOI:** 10.3390/ma15217437

**Published:** 2022-10-23

**Authors:** Yongxiang Wu, Minyi Huang, Chunlin He, Kaituo Wang, Nguyen Thi Hong Nhung, Siming Lu, Gjergj Dodbiba, Akira Otsuki, Toyohisa Fujita

**Affiliations:** 1School of Resources, Environment and Materials, Guangxi University, Nanning 530004, China; 2Zhengzhou Non-Ferrous Metals Research Institute Ltd. of CHALCO, Zhengzhou 450041, China; 3Graduate School of Engineering, The University of Tokyo, Tokyo 113-8656, Japan; 4Facultad de Ingeniería y Ciencias, Universidad Adolfo Ibáñez, Diagonal Las Torres 2640, Peñalolén, Santiago 7941169, Chile; 5Waste Science & Technology, Luleå University of Technology, SE 971 87 Luleå, Sweden

**Keywords:** nanobubbles, calcium carbonate, transformation, crystal-control, extended DLVO theory, capillary force

## Abstract

Numerous approaches have been developed to control the crystalline and morphology of calcium carbonate. In this paper, nanobubbles were studied as a novel aid for the structure transition from vaterite to calcite. The vaterite particles turned into calcite (100%) in deionized water containing nanobubbles generated by high-speed shearing after 4 h, in comparison to a mixture of vaterite (33.6%) and calcite (66.3%) by the reaction in the deionized water in the absence of nanobubbles. The nanobubbles can coagulate with calcite based on the potential energy calculated and confirmed by the extended DLVO (Derjaguin–Landau–Verwey–Overbeek) theory. According to the nanobubble bridging capillary force, nanobubbles were identified as the binder in strengthening the coagulation between calcite and vaterite and accelerated the transformation from vaterite to calcite.

## 1. Introduction

Nanobubbles are described as a gaseous domain in the liquid phase with a diameter of less than 1 μm [1] with unusual longevity lasting for several weeks or even months, which contradicts the classical Epstein–Plesset theory prediction that nanobubbles cannot exist and should disappear in a few milliseconds or microseconds [2,3]. Until now, significant progress has been made on the stability of nanobubbles [2,3,4], which has attracted the attention of experimentalists and theorists. There are more and more studies involving possible explanations for the stability of nanobubbles that have been proposed [5]. Since it was discovered that the surface of nanobubbles had a high magnitude of zeta potential, the charge stabilization model has aroused the attention of researchers [6,7,8]. In recent years, the mechanism of charge stabilization appears to be the most reasonable to rationalize nanobubble stabilization. An electrical double layer is formed on the surface of the nanobubble in water containing free moving ions, which serves as the main energy supply to withstand the coalescence of multiple bubbles and generates a repulsive pressure to balance the Laplace pressure [2].

The majority of nanobubble research has recently been rapidly increasing in many kinds of scientific disciplines. The unique physicochemical properties of the nanobubbles, such as long-term stability [2,3,6], high zeta potential [3,7], generation, and degradation of free radicals when collapsing [9], has aroused interest in many areas. Therefore, numerous reports have been published on the application of nanobubbles, including medicine [10,11], environment [9,12], flotation [13,14], and materials [15]. In medicine, nanobubbles have been used to protect proteins from surface-mediated denaturation [16] as well as for ultrasound imaging and intracellular drug delivery [17]. H_2_ nanobubbles in water were discovered to inhibit tumor cell growth [18]. The application of nanobubbles in environmental studies is well-known, such as the removal of heavy metal ions [19], wastewater treatment [20], removal of pollutants from water [21]. As mentioned above, nanobubbles are thought to have a wide range of applications. However, the application of nanobubbles in material science and engineering is uncommon, necessitating further exploration of their potential applications.

To some extent, nanobubbles can change the state and properties of water, as mentioned in many papers. Hydrogen nanobubbles in aqueous solution have a very low Eh (oxidation and reduction potential), which can be used to prevent oxidation [8]. Because a large number of hydroxyl radicals are generated when oxygen nanobubbles in aqueous solution break, they have a high oxidation state [8]. Due to their unique physicochemical properties, in this research, nanobubbles in water were used as a reaction solvent instead of pure water during calcium carbonate (CaCO_3_) synthesis by the double decomposition method to study the influence of nanobubbles on the growth of crystalline CaCO_3_.

Calcium carbonate (CaCO_3_), one of nature’s most common and widely dispersed materials, is an important component of the global carbon cycle [22]. Calcium carbonate has three crystalline phases vaterite, aragonite, and calcite [23]. In these crystals, vaterite is the least stable phase in the water, and can spontaneously change into calcite [24]. Calcite is the most common form because it is thermodynamically stable [23]. The formation of vaterite to calcite can be influenced by reaction parameters, such as supersaturation, temperature, reaction time, and additive use [25].

Furthermore, calcium carbonate has a wide range of applications that depend on its shape and morphology. Therefore, it is important to control the crystal shape or morphology of CaCO_3_ [26]. Aragonite whiskers with high aspect ratios, for example, have been in great demand for the reinforcement of polymer materials [26]. Flexible and deformable calcium carbonate is also extensively used as a filler and pigment in papermaking [27]. Surface-functionalized calcium carbonate particles can also be adapted to create novel catalytic materials [28]. Non-spherical vaterite particles are appealing as solid supports for targeted and extended drug delivery [29]. In wastewater treatment, vaterite has been explored as it is excellent at removing several heavy metal ions [30,31,32]. Porous vaterite and cubic calcite aggregated calcium carbonate are used for Cu^2+^ heavy metal removal [32]. Vaterite particles for strontium removal were also investigated and reported [31]. However, Sasamoto reported that porous calcite has potential as an adsorbent with a fast reaction rate and large adsorption amount in the removal of heavy metal ions [33].

There are several preparation methods for CaCO_3_, including the carbonation method [34], double decomposition method [35], and the thermal decomposition of calcium bicarbonate [36]. Among them, the double decomposition method is a common way to produce CaCO_3_. In this method, calcium chloride (CaCl_2_) is exploited as a calcium source, and sodium carbonate (Na_2_CO_3_) is used as a carbon source [35]. In some papers, different kinds of water-soluble additives have been used to control the crystal formation of the CaCO_3_, such as sucrose [37], anion surfactants [38], sodium dodecylbenzene sulfonate [39] and para-aminobenzoic acid [40]. Nanobubbles are clean and environmentally friendly. On the other hand, there is little research on the effect of nanobubbles on the crystal formation of calcium carbonate.

Due to the stability of the high zeta potential and the nanobubble bridging capillary force, nanobubbles could affect the crystal transition of calcium carbonate. According to previous studies [8], nanobubbles containing different gases, except CO_2_, had similar physical properties. Therefore, only air nanobubbles were applied in the present study. In future studies, CO_2_ and other gases would be discussed.

In this paper, air nanobubbles were produced using a hydraulic cavitation method [41,42]. The double decomposition method was used to produce calcium carbonate. To control the formation of CaCO_3_ crystals, nanobubbles were used instead of traditional additives in this research. The DLVO (Derjaguin–Landau–Verwey–Overbeek) theory [43,44] was used to evaluate the stability of the colloidal CaCO_3_ particles containing electrolytes in aqueous solution, while extended DLVO theory incorporating hydrophobic interaction energy [8] was used to evaluate the stability of the nanobubbles and the interaction between nanobubbles and CaCO_3_, because of the hydrophobic surface property of nanobubbles. According to the experimental results and calculations, the possible kinetics and mechanisms of nanobubbles’ influence on the transformation were proposed in detail.

## 2. Materials and Methods

### 2.1. Materials

Deionized water with a resistivity of 18.2 MΩ·cm^−1^ prepared by the Classic Water Purification System from Hitech instruments Co., Ltd. (Shanghai, China) was used for all the experiments. Calcium chloride (CaCl_2_, AR, 96.6%, Guangdong Guanghua Sci-Tech Co., Ltd., Shantou, China), and anhydrous sodium carbonate (Na_2_CO_3_, AR, 99.8%, Guangdong Guanghua Sci-Tech Co., Ltd., Shantou, China) were obtained from Guangdong Guanghua Sci-Tech Co., Ltd. (Shantou, China) and were of analytical purity. The pH of the aqueous solution was modified by the addition of hydrochloric acid (HCl, AR, 36%~38%, CHRON CHEMICALS, Chengdu, China) or sodium hydroxide (NaOH, AR, 96%, Guangdong Guanghua Sci-Tech Co., Ltd., Shantou, China).

To investigate the zeta potential of the vaterite and calcite, 100% of the calcite prepared in this work was used. 94% of the artificial vaterite was prepared by bubbling CO_2_ into 0.05 M CaCl_2_ aqueous solution, and ammonia (NH_4_OH, 30%) was used to adjust the solution pH to 10 while the solution mixture was vigorously stirred in a beaker continuously. After 15 min, the particles were rinsed with deionized water and filtered twice, and then dried in an oven at 110 °C for 0.5 h and then at 70 °C for 12 h.

The reaction of vaterite synthesis is as follows [34]:CaCl_2_ + CO_2_ + 2 NH_4_OH → CaCO_3_ + 2 NH_4_Cl + H_2_O (pH = 10)

According to the SEM image and the XRD pattern, shown in Appendix A, spherical calcium carbonate was vaterite with high purity (94%).

### 2.2. Methods

#### 2.2.1. Preparation of Nanobubble Suspension

The bulk of nanobubbles were generated by high-speed shearing using a purpose-built device (self-made equipment) which consisted of a plastic stent, an electric high-speed motor in a plastic bracket, a glass beaker, and a blade made of stainless steel as shown in Appendix A. The working principle was as follows: high-speed rotation of the blade above the motor drive creates high- and low-pressure areas and forms pressure gradients, which forms cavities in the low-pressure zones where the nanobubbles are produced [41,45].

In a typical process, the device was operated three times by adding 700 mL deionized water at 1 min intervals before producing nanobubbles. The motor frequency was initially set to 20,000 rpm. After this, 700 mL deionized water was poured into the glass beaker. The lid was put on the beaker to prevent the gas from escaping and avoid impurities in the water. The device was operated ten times, with 1 min intervals. Nanobubbles were identified after completing these steps. After generation, the solution was transferred to another glass container and sealed for the next experiment.

#### 2.2.2. Calcium Carbonate Synthesis

The flow chart of calcium carbonate synthesis is shown in Figure 1. Firstly, nanobubbles were produced, CaCl_2_ and Na_2_CO_3_ were then dissolved into the deionized water containing nanobubbles at 20 °C. The initial concentrations of CaCl_2_ and Na_2_CO_3_ in deionized water were 0.05 M (M = mol/L). 200 mL Na_2_CO_3_ solution (0.05 M) was added to 200 mL CaCl_2_ solution (0.05 M), and the solution mixture was vigorously stirred in the beaker continuously. Upon the completion of the designated reaction time (i.e., 0.5, 1, 2, 3, 4 h), the particles were rinsed with deionized water and filtered twice. After the last washing and suction filtration, the sedimented particles were dried in an oven at 110 °C for 0.5 h and then at 70 °C for 12 h before particle characterization.

#### 2.2.3. Characterization

The size of the nanobubbles was measured by the dynamic light scattering method (DLS, NanoBrook Omni, Brookhaven Instruments, Holtsville, NY, USA) and the concentration of nanobubbles was measured by the particle trajectory method (Nano-Sight, NS300, Malvern). The zeta potential measurements were made with the use of the phase analysis light scattering method (NanoBrook Omni, Brookhaven Instruments). After the synthesized CaCO_3_ particles were deposited onto a stub and coated with gold, the polymorphs and morphologies were characterized by scanning electron microscopy (SEM, SEM-EDS, Phenom ProX, Netherlands) at an accelerating voltage of 15 kV. The samples obtained in Section 2.2.2 were directly used for XRD measurements. The X-ray diffraction (XRD, Rigaku D/MAX 2500V, Japan’s neo-Confucianism) was used to verify the existence of vaterite and calcite and calculate the mass fraction of calcite within the synthesized CaCO_3_ at Cu Kα target (λ = 0.1542 nm), at 40 kV and 30 mA with a scanning speed of 8° 2θ/min. The CaCO_3_ particles with potassium bromide tableting were measured by Fourier transform infrared spectroscopy (FT-IR, Nicolet iS 50, Thermo Scientific) to verify the existence of vaterite and calcite using the transmission mode with a scanning speed of 20 spectra/s. Typical spectral resolution was 0.25 cm^−1^.

To quantify different crystalline calcium carbonates and calculate the mass fraction of calcite within the synthesized CaCO_3_, the intensities of the (110), (112), and (114) crystallographic planes for vaterite (I_110V_), (I_112V_), and (I_114V_), and the (104) crystallographic plane for calcite (I_104C_), the semi-quantitative phase compositions were calculated by using Equations (1) and (2) [46]:(1)$$X_{v}=(I_{110v}+I_{112v}+I_{114v})/(I_{104c}+I_{110v}+I_{110v}+I_{110v})$$
(2)$$X_{c}=1-X_{v}$$
where X_v_ is the mass fraction of vaterite, X_c_ is the mass fraction of calcite, I_104C_ is the intensities of the (104) crystallographic plane for calcite, and I_110v_, I_112v_ and I_114v_ are the intensities of the (110), (112), and (114) crystallographic planes for vaterite, respectively.

## 3. Results

### 3.1. Generation of Air Nanobubbles

Because the whole reactor was sealed, impurities were unlikely to be found in this solution. Furthermore, the content of impurities in water was extremely low, indicating that the colloid particles observed in the solution were nanobubbles. Due to the Tyndall effect based on the light scattering of colloidal sized objects being illuminated with a laser beam after high-speed shearing, a clear light path was observed in the solution, demonstrating that plenty of colloid particles were produced. In this experiment, the hydrodynamic diameter of the air nanobubbles was measured by DLS. The mean diameter of the nanobubbles was about 83.6 nm in the absence of any salt addition. As shown in Table 1, in the 0.05 M CaCl_2_ and 0.05 M Na_2_CO_3_ nanobubble aqueous solutions, the mean diameter was 126.2 nm and 101.8 nm, respectively. The pH of the Na_2_CO_3_ aqueous solution was 11.7, while the pH of the deionized water and CaCl_2_ aqueous solution was about 6. As Ca^2+^ tended to be adsorb onto the surface of nanobubbles in a CaCl_2_ aqueous solution, the nanobubbles’ zeta potential was positive (7.3 mV). As shown in Figure 2, the concentration of nanobubbles in deionized water was 1.6 × 10^6^, which was much lower than the prepared nanobubble water. According to previous studies [2,43], the mixing of organic solvents with pure water leads to the spontaneous formation of nanobubbles, while nanobubbles cannot form spontaneously in deionized water. the concentration of the bulk of the nanobubbles shifted to the right in the 0.05 M CaCl_2_ and 0.05 M Na_2_CO_3_ solutions, which indicated that their mean diameters increased. Simultaneously, in the presence of a strong electrolyte, the mono-dispersity and the total concentration of the bulk nanobubbles reduced, especially noticeable in the 0.05 M CaCl_2_ solution.

### 3.2. The Influence of Nanobubbles on the Transformation from Vaterite to Calcite

To study the influence of nanobubbles on the transformation from vaterite to calcite, deionized water or nanobubble water was used as a solvent. The products obtained with different reaction times in the presence/absence of nanobubbles were characterized by XRD. The XRD patterns of the obtained samples with different reaction times at 25 °C in nanobubble water are given in Figure 3a. In comparison with their standard JCPDS files (vaterite, 720506) and (calcite, 830577), at reaction times 0.5, 1, 2, and 3 h, a mixture of calcite and vaterite was identified. The peaks corresponding to vaterite vanished as the reaction time progressed, while the peak at 2θ = 29.46 corresponding to calcite became sharper and higher. After 4 h, pure calcite was identified. However, the XRD patterns of the obtained samples with different reaction times in deionized water in Figure 3b indicated that the peaks corresponding to calcite rapidly increased in intensity as vaterite transformed into calcite, but after 4 h, a mixture of calcite and vaterite was still identified.

Comparing Figure 3a,b, when the deionized water containing nanobubbles was used as a solvent, the transformation from vaterite to calcite was completed in less time, indicating that nanobubbles could enhance this transformation.

According to the XRD patterns, the phase compositions calculated from Equations (1) and (2) are shown in Figure 4, which demonstrates the change in the mass fraction of calcite. Regardless of whether pure water or nanobubble water was used as the solvent, the vaterite mass fraction decreased and vanished with reaction time. After 1 h, the rate of transformation from vaterite to calcite decreased, and this might be because the nanobubbles gradually decreased with the progress of the reaction, and their effect on the transformation became weaker. After 4 h, when nanobubble water was used as the solvent, all the vaterite was converted to calcite. However, when deionized water was used as the solvent, a mixture of calcite and vaterite remained, with the mass of calcite accounting for only about 60% after 4 h, and took more than 6 h to finish this transformation from vaterite to calcite. The transformation from vaterite to calcite is a spontaneous process. The driving force for the acceleratory transformation by the present of nanobubbles was the vaterite–nanobubble coagulation. This is described in the following discussion Section 4.

To confirm whether vaterite remains in the products as the reaction progresses, FT-IR spectra of CaCO_3_ were obtained after 4 h in the nanobubble water and deionized water and are plotted in Figure 5. The absorption peaks at 709 cm^−1^ indicate the presence of calcite and the peak at 748 cm^−1^ verifies the presence of vaterite [46,47,48]. Figure 5 indicates the presence of calcite in deionized water containing nanobubbles (a) while the presence of a mixture of calcite and vaterite in deionized water (b), and these results were in good accordance with the XRD results shown in Figure 3. The absorption peak at 709 cm^−1^ (in-plane vibration) of calcite (b) obtained in deionized water is slightly red shifted, compared with that of sample (a). The phase transformation from vaterite to calcite might have caused the peak shift due to the slight change in the atomic distance [49].

SEM images of the fabricated samples are shown in Figure 6, with different reaction times in nanobubble water (a) to (d) and deionized water in the absence of nanobubbles (e) to (h). In deionized water containing nanobubbles, a mixture of spherical particles, i.e., crystals of vaterite [22,25,50], and cubic crystals, i.e., crystals of calcite [22,25,50], were found as shown in Figure 6a–c. It was obvious that the number of cubic crystals in deionized water containing nanobubbles increased over time with only cubic crystals present after 4 h, as shown in Figure 6d. On the other hand, in deionized water without nanobubbles, all of the products obtained at different reaction times were a mixture of spherical particles and cubic particles as shown in Figure 6e–h. The results obtained by SEM images agreed well with the results obtained by XRD (Figure 3 and Figure 4) and FT-IR (Figure 5).

### 3.3. The Influence of Concentration of Nanobubbles on the Transformation from Vaterite to Calcite

To study the effect of nanobubbles on the polymorphs of CaCO_3_, deionized water with different concentrations of nanobubbles was used as a solvent. In this experiment, high-speed shearing was used to produce nanobubbles. The same motor frequency and shearing time were chosen to avoid multiple variables occurring at the same time.

The volume ratio of deionized water to nanobubbles in groups 1 to 4 is listed in Table 2. The low concentration of nanobubbles in deionized water (Groups 2 and 3) was prepared by adding deionized water containing nanobubbles obtained by high-speed shearing in different volumes to the deionized water. In groups 2 and 3, the volume ratio of pure water to nanobubbles (V_deionized water containing nanobubble_: V_deionized water_) was 1:3 and 1:1, respectively. In group 1, only deionized water was used as a solvent, while in group 4, nanobubble water without dilution was used. The total nanobubble number was from 0 to 2.96 × 10^8^ bubbles/mL.

The XRD patterns of the obtained samples with different concentrations of nanobubbles in deionized water are shown in Figure 7. When the concentration of nanobubbles (groups 1 to 3) were low, mixtures composed of vaterite and calcite were obtained. When the concentration of nanobubbles was 2.96 × 10^8^ bubbles/mL (group 4), only peaks corresponding to calcite were found.

According to the calculations using Equations (1) and (2), the mass fraction of the calcite (W_calcite_/W_calcite+vaterite_) as a function of the concentration of nanobubbles in deionized water after 4 h is shown in Figure 8. As the concentration of nanobubbles increased, the mass fraction of calcite increased linearly and reached 100% when the concentration of nanobubbles was at 2.96 × 10^8^ bubbles/mL.

SEM images of CaCO_3_ obtained after 4 h in groups 1–4 are shown in Figure 9a–d, respectively. In Figure 9a–c, the products obtained were a mixture of spherical vaterite particles and cubic calcite particles, while in deionized water containing a high concentration of nanobubbles (2.96 × 10^8^ bubbles/mL), only cubic calcite particles were observed, as shown in Figure 8. The results obtained by SEM images were quite consistent with those obtained by XRD, as shown in Figure 7. The size of calcite and vaterite particles was almost comparable in deionized water containing various concentrations of nanobubbles. The size distribution of calcite after 4 h prepared in 2.96 × 10^8^ bubbles/mL of nanobubble water is shown in Appendix A. The mean diameter of calcite was around 5 μm.

## 4. Discussion

### 4.1. CaCO_3_ Particle Synthesis with Different Nanobubble Concentrations

The reaction of CaCO_3_ synthesis is as follows:Na_2_CO_3_ + CaCl_2_ → CaCO_3_ + 2NaCl(3)

During this reaction, 0.05 M of Na_2_CO_3_ aqueous solution was reacted with an equal amount of CaCl_2_ aqueous solution of the same concentration, the synthesized CaCO_3_ was 0.025 M (C_1_). One particle diameter (d) of synthesized CaCO_3_ particle was around 5 µm as shown in Appendix A. The average density of CaCO_3_ (ρ) was about 2.7 g/mL (calcite 2.71, vaterite 2.65 [51]). The total number of CaCO_3_ particles synthesized (N_CaCO3_) is as follows,
(4)$$N_{{\mathrm{CaCO}}_{3}}=V/V_{{\mathrm{CaCO}}_{3}}=6M_{{\mathrm{CaCO}}_{3}}\times C_{1}\times V_{\mathrm{solution}}/\left({\pi \rho d}^{3}\right)$$

The concentration of CaCO_3_ particles (C, particles/mL) is given by Equation (5),
(5)$$C=N_{{\mathrm{CaCO}}_{3}}/V_{\mathrm{solution}}=6M_{{\mathrm{CaCO}}_{3}}\times C_{1}/\left({\pi \rho d}^{3}\right)=1.4\times 10^{7}\mathrm{particles}/\mathrm{mL}$$
where V is the total volume of calcium carbonate obtained, $V_{{\mathrm{CaCO}}_{3}}$ is the volume of a single calcium carbonate particle, $M_{{\mathrm{CaCO}}_{3}}$ is the molecular weight of calcium carbonate, V_solution_ is the volume of the solution, and d is the mean diameter of the calcium carbonate particles.

As the nanobubble number was 2 to 3 × 10^8^ bubbles/mL as shown in Table 2, the number of nanobubbles applied for this experiment was 10 times as many as that of CaCO_3_ particles.

The interaction between nanobubbles and CaCO_3_ particles is important for the preparation of calcite and vaterite. After the reaction shown in Equation (3), the pH equilibrated around 10. Figure 10 shows the zeta potential of air nanobubbles, calcite, and vaterite in the 0.05 M of NaCl aqueous solution containing different concentrations of Ca^2+^ at pH = 10. At pH = 10, the zeta potential of vaterite was always positive, while calcite and air nanobubbles were negative at concentrations less than 10^−3^ M of Ca^2+^.

After the reaction described in Equation (3), the Ca^2+^ concentration in the solution was constant at 2.5 × 10^−5^ M after 1.5 h, as shown in Appendix A. Therefore, after 1.5 h the zeta potential of calcite and nanobubbles was negative and the zeta potential of vaterite was positive, as shown in Figure 10.

### 4.2. The Interaction between Nanobubbles, CaCO_3_ Particles and Nanobubble-CaCO_3_ Particles

In this section, we evaluate (a) bubble–bubble interaction, (b) vaterite–calcite interaction, (c) bubble–calcite interaction, and (d) bubble–vaterite interaction, by using the classical and extended DLVO theory in order to investigate and propose the mechanism of the transformation from vaterite to calcite in the presence of air nanobubbles.

In addition to the classical DLVO theory, i.e., the summation of the attractive van der Waals interaction energy (E_A_) and the repulsive electrostatic interaction energy (E_R_), to evaluate the interactions between different CaCO_3_ particles (vaterite-calcite), we used the hydrophobic interaction energy (E_h_) to evaluate the stability of the nanobubbles. The total potential energy (E_T_) between two nanobubbles is described as follows [8,52,53]:E_T_ = E_A_ + E_R_ + E_h_(6)
(7)$$\frac{E_{T}}{\mathrm{kT}}=\frac{E_{A}+E_{R}+E_{h}}{\mathrm{kT}}$$

The total potential energy (E_T_) is a key indicator of the stability of nanobubbles and can evaluate whether the nanobubbles coagulate or disperse. When E_T_ is significantly positive (e.g., >15 kT), there is a potential barrier between the two adjacent bubbles which will not coagulate, while if E_T_ is slightly positive or negative (<15 kT), the nanobubbles can coagulate together and form a larger bubble.

E_A_ + E_h_ and E_R_ can be described by the following formulas:(8)$$E_{A}+E_{h}=-\frac{A+K}{6}\{\frac{2a_{1}a_{2}}{R^{2}-{\left(a_{1}+a_{2}\right)}^{2}}+\frac{2a_{1}a_{2}}{R^{2}-{\left(a_{1}-a_{2}\right)}^{2}}+\ln\left[R^{2}-{\left(a_{1}+a_{2}\right)}^{2}\right]-\ln[R^{2}-{\left(a_{1}-a_{2}\right)}^{2}]\}$$
(9)$$R=a_{1}+a_{2}+h$$
(10)$$E_{R}=\frac{{\pi \varepsilon }_{r}\varepsilon_{0}a_{1}a_{2}\left(\psi_{1}^{2}+\psi_{2}^{2}\right)}{a_{1}+{\;a}_{2}}\{\frac{2\psi_{1}\psi_{2}}{\psi_{1}^{2}+\psi_{2}^{2}}\ln\frac{1+\exp\left(-\kappa h\right)}{1-\exp\left(-\kappa h\right)}+\ln[1+\exp\left(-2\kappa h\right)]\}\;$$
where A is the Hamaker constant, K is the hydrophobic force constant, a_1_ and a_2_ are the radii of the nanobubbles, h is the surface distance of the nanobubbles, κ represents the Debye–Hückel parameter, and ψ_1_ and ψ_2_ represent the surface potential of the nanobubbles of radii a_1_ and a_2_, respectively. The thickness of the electric double layer can be represented by the Debye length (λ_D_ = 1/κ), which can be described by the following formula:(11)$$\kappa =\sqrt{\frac{2{\mathrm{nz}}^{2}e^{2}}{\varepsilon_{r}\varepsilon_{0}\mathrm{kT}}}$$
n = 1000N_A_C(12)
where n is the concentration of anions or cations in the solution, z represents the valence of an ion, e represents the electron charge, ε_r_ represents the relative dielectric constant, ε_0_ represents the permittivity of a vacuum, T represents the temperature (K), N_A_ represents the Avogadro number and C represents concentration of anions or cations (mol/L). 

Bubble–bubble interaction: according to our previous study [8], these nanobubbles could stay stable for several weeks or even months. Even in the presence of salt, nanobubbles can also be stable for days. They were stable enough to perform zeta potential measurements. The zeta potential of nanobubbles is shown in Figure 10. When the nanobubbles existed in the reacted solution with a Ca^2+^ concentration less than 10^−3^ M, the zeta potential of the nanobubble was negative, as shown in Figure 11. The mean diameter of the nanobubbles in the mixed vaterite and calcite suspension after reaction depending on the time measured by DLS after filtration through 10–15 μm filter paper is shown in Figure 11. The mean diameter of the nanobubbles increased from an initial 100–130 nm (Table 1) to 600–700 nm during 3 h of the reaction time (Figure 11). Figure 12a shows the total potential energy (E_T_) between the bubbles calculated by using the extended DLVO theory, and there was no potential barrier to prevent the coalescence of the bubbles, which was consistent with the bubble size enlargement as shown in Figure 11.

Vaterite–calcite interaction: Figure 12b shows the total potential energy (E_T_) between the calcite and vaterite particles (6 µm mean diameter as shown in Appendix A) calculated by using the classical DLVO theory as this was an interaction between solid particles. There was no potential barrier when considering the low zeta potential of calcite (−10 mV) and vaterite (20 mV) indicating their coagulation, (Figure 10).

Air bubble–calcite interaction: Figure 13 shows the total potential energy (E_T_) between the different sizes of air bubbles and calcite particles as a function of their distance by the extended DLVO theory. Figure 13a,b depict the interactions between the initial 120 nm and 700 nm nanobubbles after 1 h (shown in Figure 11) and the prepared 5 μm calcite after 1 h (shown in Appendix A), respectively. In the both cases, there was no potential barrier between nanobubbles and calcite, indicating their hetero-coagulation.

Air bubble–vaterite interaction: considering the zeta potential of the vaterite particles was positive (20 mV) and air bubbles was negative, as shown in Figure 13c,d, there was no potential barrier between the nanobubbles and vaterite indicating hetero-coagulation between nanobubbles and vaterite.

To sum up, in deionized water containing nanobubbles, according to the calculations based on the extended DLVO theory, there was no potential barrier between nanobubbles, calcite or vaterite, indicating that negatively charged nanobubbles could attach and accumulate not only on the surfaces of calcite particles but also on the surfaces of vaterite particles. This supported the premise that nanobubbles could bind calcite with vaterite.

### 4.3. The Capillary Force Model between Calcite and Vaterite

Figure 14 shows SEM images of spherical vaterite particles attached to the cubic surface of calcite particles in deionized water (a) and nanobubble-containing solution (b) after 0.5 h. Obviously, in the nanobubble-containing solution (b), there were more spherical vaterite particles attached to the surface of cubic calcite particles than that in the deionized water without nanobubbles (a). The zeta potential of the vaterite particles was positive, while the surface of the calcite particles was negative. The vaterite particles tended to be clustered with calcite particles. Furthermore, compared to vaterite particles, calcite particles were cubic crystals, which had a higher probability of coming into contact with vaterite particles. This indicates that nanobubbles can bind calcite with vaterite, enabling vaterite and calcite to clustered together in deionized water containing nanobubbles.

The capillary force model [54,55,56,57], based on the forces caused by the pressure drop across the interface at the neck and the interfacial tension force, was used to calculate the capillary force between the vaterite and calcite particles:(13)$$F_{c}=\pi \sigma \sin\alpha [2\sin\left(\alpha +\theta \right)+\mathrm{Rsin}\alpha \left(1/r-1/l\right)$$
(14)r=[R(1−cosα)+h]/[cos(α+θ)+cosα]
(15)l=Rsinα−r[1−sin(α+θ)]
where F_c_ is the capillary force, σ is the surface tension coefficient of the liquid, α is the fill angle, R is the radius of the colloidal probe, θ is the liquid contact angle, r and l are the principal radii of the capillary bridge, and h is the separation distance between the vaterite and calcite surfaces, as shown in Figure 15a. 

In absence of nanobubbles, spherical vaterite attached to the surface of cubic calcite might become separated by the repeated collisions between solid particles and high-speed flowing solution stirred under the magnetic agitator [58]. It can be difficult for ultrafine particles to collide and coagulated. On the other hand, in presence of nanobubbles, the negatively charged nanobubbles can attach and accumulate on the surface of calcite particles and this aggregate can further coagulate with positively charged vaterite particles [58].

According to previous studies, the probability of collision between a particle and a bubble is closely related to bubble size [59]. Because of the smaller contact area and aggregation effects on ultrafine particles, nanobubbles are more easily attached to the surface of vaterite and calcite particles [13] in comparison with larger bubbles. According to the nanobubble bridging capillary force (NBCF) resulting from the long range hydrophobic attractive force [55,60,61,62], as they approached, a gaseous capillary bridge could be formed through the coalescence of nanobubbles. The underlying mechanism is described in Figure 15b. 

The induced attractive capillary force was helpful to enhance the aggregation of vaterite and calcite particles through bubble bridging. The nanobubble bridging capillary force as a function of the surface separation distance between vaterite and calcite particles is shown in Figure 15c. Even if the separation distance was 1000 nm, F_c_ was still 0.001 mN/m, which would avoid separation of concentrated solid particles and enhance the effective slip lengths between solid and liquid, enhancing the aggregation between solid particles [63]. Once vaterite and calcite particles aggregated together, the capillary forces produced in the presence of the nanobubbles prevented them from separating. The calculation results agreed well with the results obtained by SEM images (Figure 14). There were more spherical vaterite particles attached to the surfaces of cubic calcite particles in deionized water in the presence of nanobubbles than that in deionized water in the absence of nanobubbles.

### 4.4. Formation Mechanism of the Crystallization Transformation of CaCO_3_

In this series of experiments reported in this article, the double decomposition method was used to produce calcium carbonate. After mixing CaCl_2_ and Na_2_CO_3_ solution, amorphous calcium carbonate (ACC) formed in several seconds, then ACC transformed into vaterite, and finally, all of them changed into calcite. This transformation occurred in two stages: the first step was the transformation from ACC to vaterite, and the second step was the rate-determining process from vaterite to calcite. Gas–water interfaces might act as nucleation sites, which would enhance the transformation from ACC to vaterite. According to the literature [22,23,64], the second stage of the reaction is much slower than the first stage. However, the second step directly determines the formation of calcite. Through the dissolution–reprecipitation mechanism, vaterite can be transformed into calcite [22,23,64]. For detail, the process was divided into two steps, the first step was the dissolution of vaterite, and the second step was the growth of the calcite, which was the rate-determining process. The second step was the recrystallization of calcium carbonate which was divided into the dissolution of vaterite and the growth of calcite [22,50,64]. As shown in Figure 16, Vaterite can dissolve in water and release calcium and carbonate ions, which are then reprecipitated on the surfaces of the growing calcite crystals. As shown in Figure 4 and Figure 8, after 4 h, all the vaterite was converted to calcite in deionized water containing nanobubbles, while, in absence of nanobubbles, a mixture of calcite (60%) and vaterite (40%) remained. With the concentration of nanobubbles increasing, the mass fraction of calcite increased linearly and reached 100% when the concentration of nanobubbles was at 2.96 × 10^8^ bubbles/mL. The results show that nanobubbles can enhance the transformation from vaterite to calcite. According to the calculation shown in Figure 13, there was no potential barrier between nanobubble, calcite and vaterite, indicating hetero-coagulation between them. The number of nanobubbles applied for this experiment was 10 times as many as that of CaCO_3_ particles. Under the intense stirring of the magnetic agitator, nanobubbles improved the probability of collision. Due to the nanobubble bridging capillary force [63] increases with shorter separation distances between the vaterite and calcite particles, as shown in Figure 15c; thus, their aggregation can become more stable. As clearly shown in Figure 14, it was easier and more stable for vaterite and calcite to become aggregated together in the presence of nanobubbles (a) compared with the absence of nanobubbles (b). This aggregation allowed more Ca^2+^ to attach to the surface of calcite, allowing calcite to grow more efficiently and transform faster.

## 5. Conclusions

This was the first study using nanobubbles to aid the transformation of vaterite to calcite, and we confirmed that the transformation was accelerated in the presence of air nanobubbles. According to the calculations based on the extended DLVO theory, in the reacted solution, there was no potential barrier between the nanobubbles and both calcite or vaterite particles, indicating their coagulation. In deionized water without nanobubbles, spherical vaterite attached on the surface of cubic calcite particles of about 5 μm in size but would be dispersed by the repeated collisions between solid particles and high-speed flowing solution. However, according to the calculations of the nanobubble bridging capillary force and the SEM images of synthesized calcite particles, nanobubbles played an important role as the binder in strengthening the coagulation between calcite and vaterite. Therefore, the transformation rate increased in deionized water containing nanobubbles, compared with the aqueous solution without nanobubbles. Based on the results, this research provides a novel method for controlling the crystal morphology and reaction kinetics of CaCO_3_. The CO_2_ bubbling method (indirect carbonation) is an important way to synthesize vaterite. The size of the bubbles has an obvious effect on the vaterite content and can be controlled by adjusting the ventilation equipment. In future studies, the influence of the bubble size on crystal formation of calcium carbonate should be studied to produce pure vaterite.

## Figures and Tables

**Figure 1 materials-15-07437-f001:**
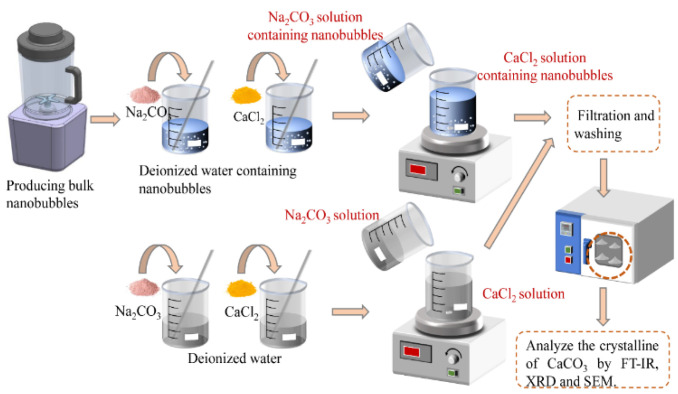
The flow chart of the synthesis of CaCO_3_.

**Figure 2 materials-15-07437-f002:**
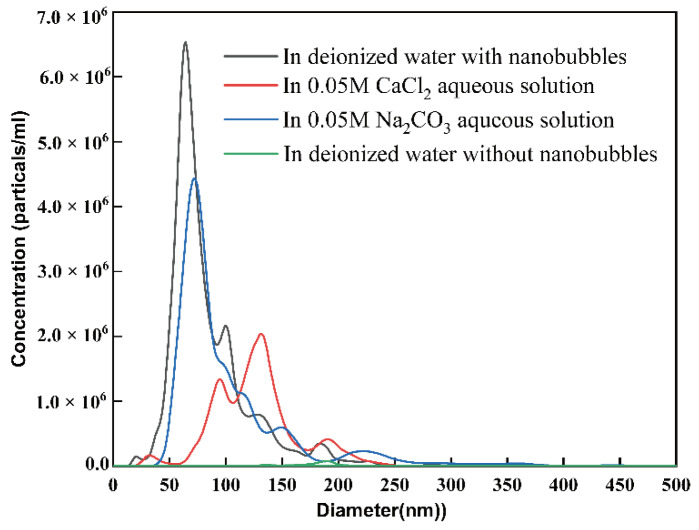
Size distribution of nanobubbles in deionized water with nanobubbles, in 0.05 M CaCl_2_ aqueous solution, 0.05 M Na_2_CO_3_ aqueous solution or in deionized water without nanobubbles.

**Figure 3 materials-15-07437-f003:**
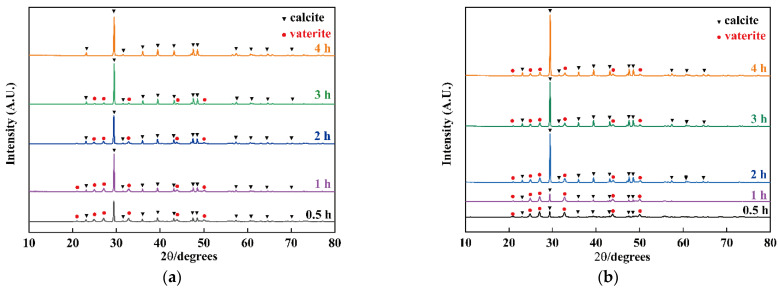
XRD patterns of CaCO_3_ particles obtained with different reaction times in deionized water containing nanobubbles (**a**) and deionized water (**b**).

**Figure 4 materials-15-07437-f004:**
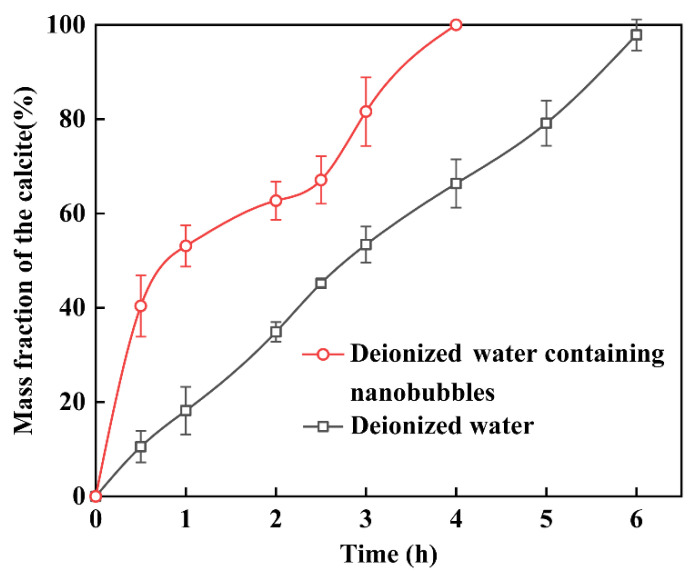
Mass fraction of the calcite (W_calcite_/W_calcite+vaterite_) as a function of the reaction time in deionized water containing nanobubbles and deionized water.

**Figure 5 materials-15-07437-f005:**
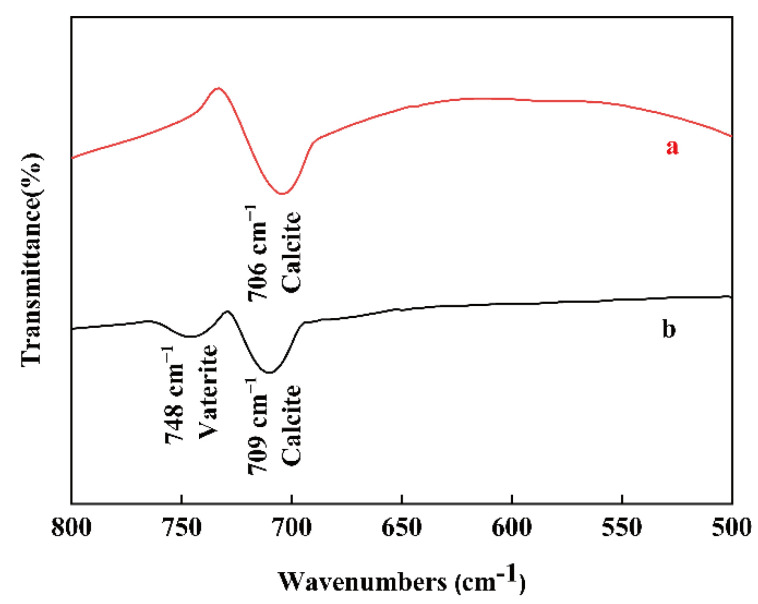
FT-IR spectra of CaCO_3_ obtained after 4 h in deionized water containing nanobubbles (a) and deionized water (b).

**Figure 6 materials-15-07437-f006:**
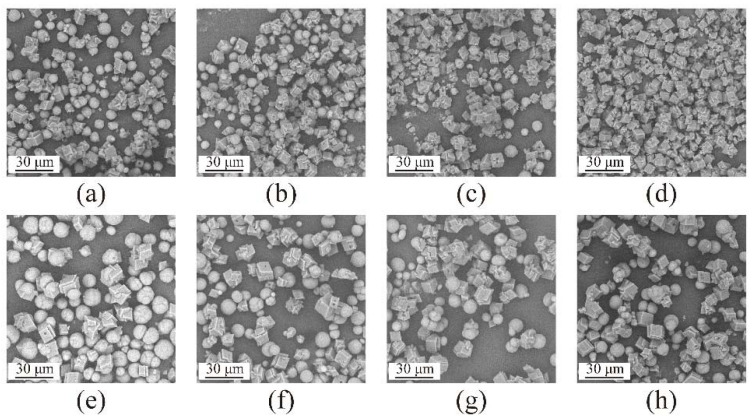
SEM images of CaCO_3_ obtained in deionized water containing nanobubbles at different reaction times 1 h (**a**), 2 h (**b**), 3 h (**c**), and 4 h (**d**) and in deionized water at different reaction times 1 h (**e**), 2 h (**f**), 3 h (**g**) and 4 h (**h**).

**Figure 7 materials-15-07437-f007:**
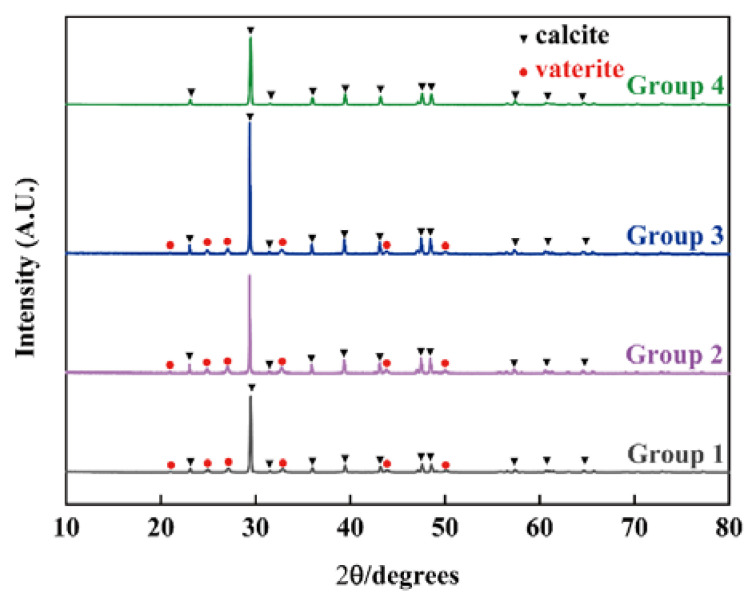
XRD patterns of CaCO_3_ particles obtained after 4 h as a function of the concentration of nanobubbles in deionized water in groups 1 to 4.

**Figure 8 materials-15-07437-f008:**
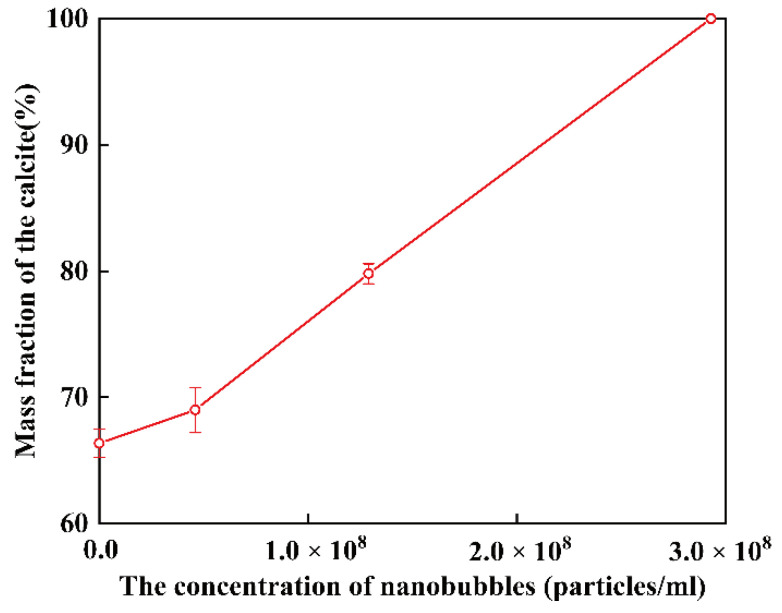
Mass fraction of the calcite (W_calcite_/W_calcite+vaterite_) as a function of the concentration of nanobubbles in deionized water after 4 h.

**Figure 9 materials-15-07437-f009:**
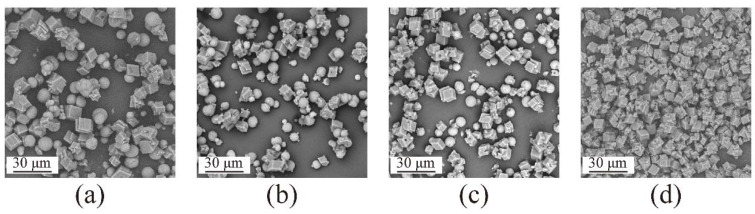
Mass fraction of the calcite (W_calcite_/W_calcite+vaterite_) as a function of the concentration of nanobubbles in deionized water after 4 h, group 1 (**a**), group 2 (**b**), group 3 (**c**), group 4 (**d**).

**Figure 10 materials-15-07437-f010:**
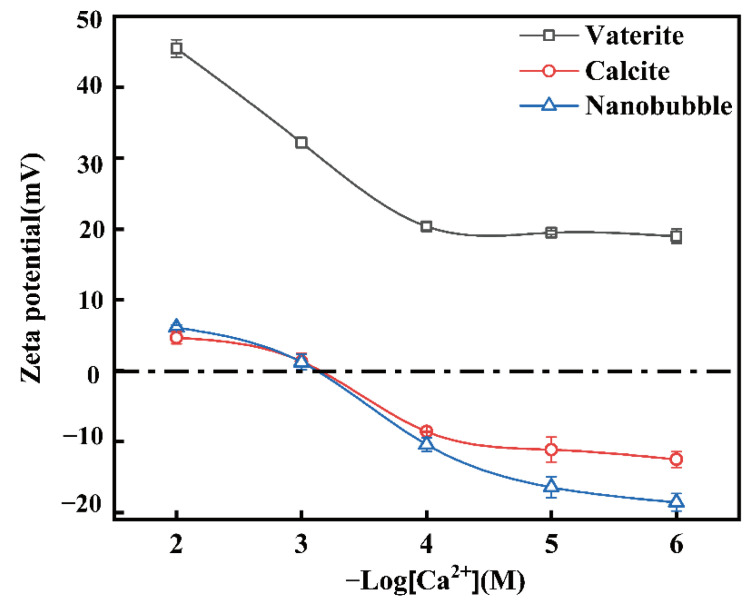
Zeta potential of air nanobubble, calcite, and vaterite in a 0.05 M of NaCl aqueous solution containing various concentrations of Ca^2+^ at pH = 10.

**Figure 11 materials-15-07437-f011:**
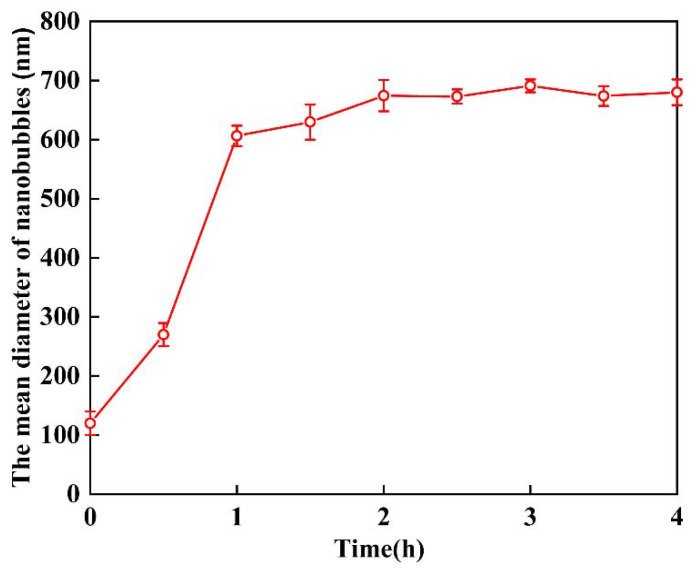
The mean diameter of nanobubbles in mixed vaterite and calcite suspension after reaction depending on the time measured by DLS after filtration with 10–15 μm filter paper.

**Figure 12 materials-15-07437-f012:**
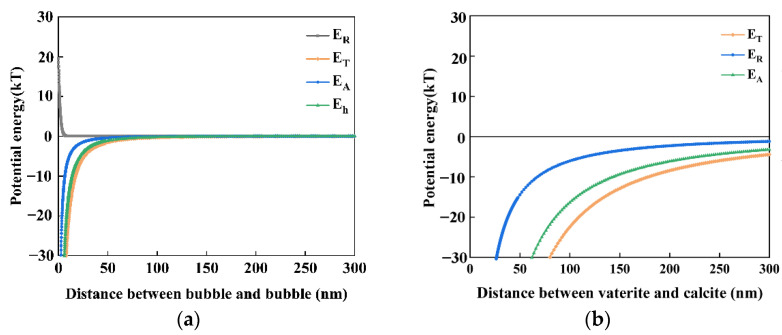
(**a**) Total potential energy E_T_ between bubbles (700 nm, −15 mV) as a function of the distance between two bubbles by the extended DLVO theory, at a Na^+^ concertation of 0.05 M and a Ca^2+^ concentration of 2 × 10^−5^ M; (**b**) Total potential energy E_T_ between calcite (3 μm, −10 mV) and vaterite (6 μm, 20 mV) as a function of the distance between two particles by the classical DLVO theory, at a Ca^2+^ concentration of 2.5 × 10^−5^ M.

**Figure 13 materials-15-07437-f013:**
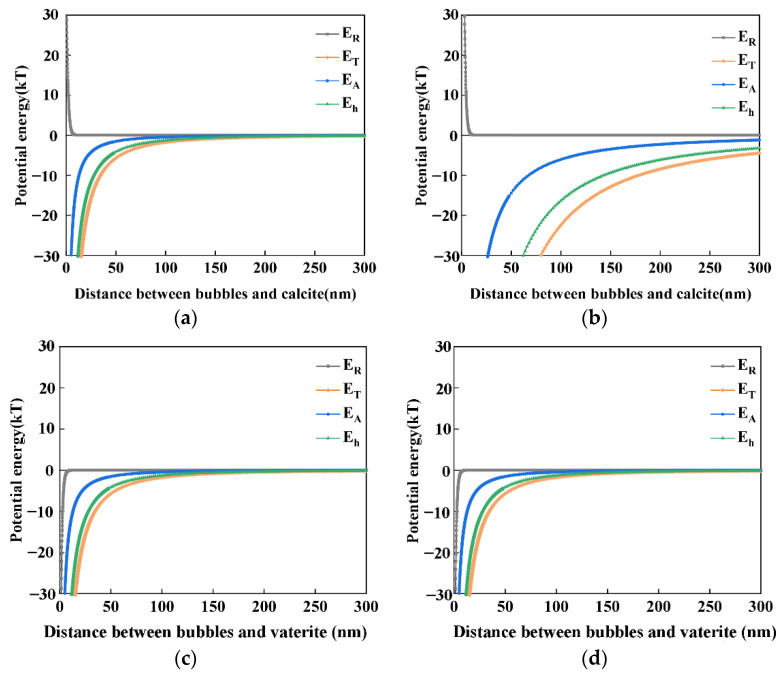
Total potential energy E_T_ between the different sizes of air bubbles and calcium carbonate particles as a function of the distance between a nanobubble and a calcium carbonate particle by the extended DLVO theory (K = 10^−19^ J). Na^+^ concentration was 0.05 M and Ca^2+^ concentration was 2 × 10^−5^ M. (**a**) Air nanobubbles (120 nm, −15 mV) and calcite particles (5 μm, −10 mV); (**b**) Air nanobubbles (700 nm, −15 mV) and calcite particle (5 μm, −10 mV); (**c**) Air nanobubbles (120 nm, −15 mV) and vaterite particles (6 μm, 20 mV); (**d**) Air nanobubbles (700 nm, −15 mV) and vaterite particles (6 μm, 20 mV).

**Figure 14 materials-15-07437-f014:**
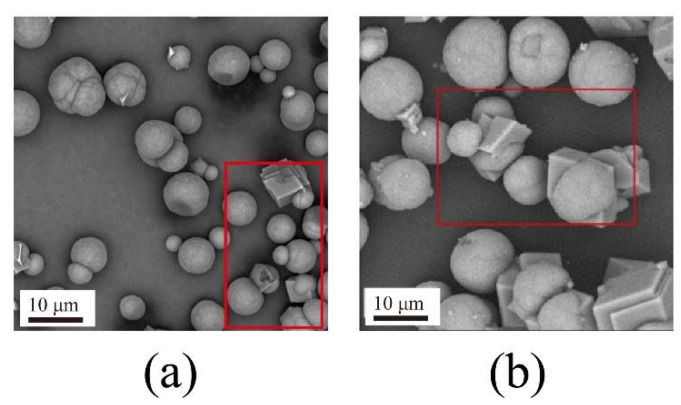
SEM images of spherical vaterite particles attached onto cubic calcite particle surfaces in deionized water (**a**) and in deionized water containing nanobubbles (**b**) (after 0.5 h passed).

**Figure 15 materials-15-07437-f015:**
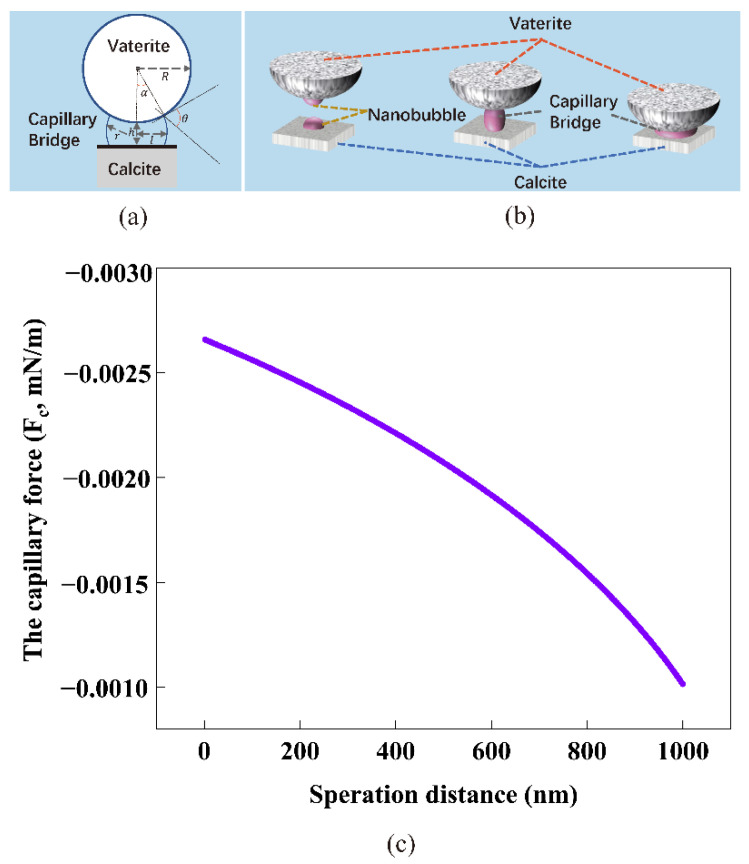
The geometry of vaterite and calcite particles used to calculate the capillary force between them (**a**); simplified mechanism of a nanobubble bridging the surfaces of vaterite and calcite particles (**b**); the nanobubble capillary force bridging vaterite and calcite particles as a function of their surface distance (**c**).

**Figure 16 materials-15-07437-f016:**
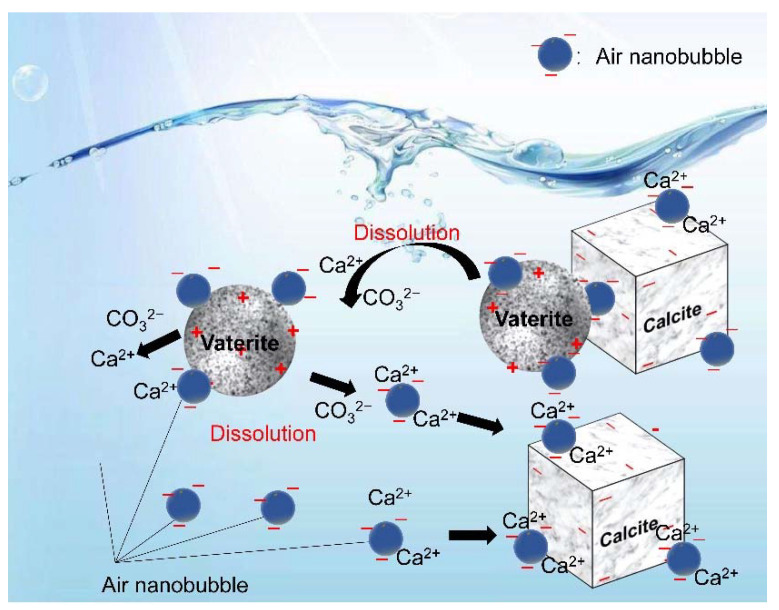
The interaction between nanobubble, vaterite, and calcite and transformation from vaterite to calcite. (Ca^2+^ concentration 10^−4^ to 10^−6^ M, pH = 10, vaterite mean diameter 5 μm, air nanobubble 120 nm.).

**Table 1 materials-15-07437-t001:** Mean diameter, zeta potential, and concentration of nanobubbles in deionized water, in 0.05 M CaCl_2_ aqueous solution or 0.05 M Na_2_CO_3_ aqueous solution.

	Mean Diameter (nm)	Zeta Potential(mV)	Concentration (bubbles/mL)	pH
In deionized water	83.6	−23.0	2.96 × 10^8^	6.3
In 0.05 M CaCl_2_ aqueous solution	126.2	7.3	1.48 × 10^8^	5.6
In 0.05 M Na_2_CO_3_ aqueous solution	101.8	−17.8	2.49 × 10^8^	11.7

**Table 2 materials-15-07437-t002:** The concentration of nanobubbles in deionized water in groups 1 to 4.

	The Volume Ratio of Pure Water to Nanobubbles (V_deionized water containing nanobubble_: V_deionized water_)	The Concentration of Nanobubbles (Bubbles/mL)
Group1	Deionized water	0
Group2	1:3	4.59 × 10^7^
Group3	1:1	1.29 × 10^8^
Group4	Initial deionized water containing nanobubbles	2.96 × 10^8^

## Data Availability

The authors confirm that the data supporting the findings of this study are available within the article.

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
