# Peer review of "The Influence of Air Nanobubbles on Controlling the Synthesis of Calcium Carbonate Crystals"

_materials, 2022, doi:10.3390/ma15217437_

Round 1
Reviewer 1 Report
This manuscript investigates the the influence of air nanobubbles on controlling the synthesis of calcium carbonate crystals. It is very well written and the results are presented in a very comprehensive way. I have only some questions that I can't answer when reading the manuscript:
My first question is the following: the authors tried to correlate the zeta potential of nanobubbles with the zeta potential of calcium carbonate crystals and their concentration. How stable are these nanobubbles? Are they stable enough for the experimenter to perform the zeta potential measurement?
In several recent studies done in batch reactors and in microchips,it was shown that vaterite, aragonite and calcite may be stabilized at an oil-water interface due to lower interfacial tension and decrease of the required Gibbs free energy. There, oil-water interfaces act as nucleation sites. I am wondering if the nanobubbles are the nucleation sites and if they stabilize more easily calcite or they favor transformation of vaterite to calcite.
I see many SEM images with vaterite and calcite be connected. I can understand the theory provided by the authors, however, how certain are they that there are two crystals vaterite and calcite connected and it is not 2 vaterite crystals one of which has already been transformed to calcite?
Yours sincerely,
Reviewer 2 Report
This manuscript deals with the influence of air nanobubbles on controlling the synthesis of calcium carbonate crystals.
In my point of view, this manuscript discloses a potential strategy for synthesizing calcium carbonate crystals. The manuscript is well organized and would be of interest to readers of Materials. I would recommend this manuscript for a minor revision.
Some comments are as follows:
1) In the experimental part, the authors explained briefly information details about XRD and FTIR however, how are the samples prepared before the measurement in each technique? The authors could specify X-ray source tube, voltage, current and scan speed used during measurement for XRD. Also, for the FTIR measurement, the authors could include the scanning mode, namely,
ATR or transmission mode and scan speed. Please specify in more detail because the researchers need to repeat and/or follow your report.
2) Results and discussions, Figure on page 7, the slightly red shift in the FTIR spectrum of sample (b) is observed when compared with that of sample (a). The peak positions are not the same wavenumber at 709 cm-1. Do the authors have an explanation for the phenomena? Please explain more about the possible reason!
Reviewer 3 Report
The MS by Fujita et al. presents an interesting structural transition of vaterite to calcite in the presence of air nanobubbles. The MS fully fills the scope of Materials and can be published after taking into consideration these points:
1) What is the driving force of vaterite to calcite transformation? According to Fig 4 the process proceeds in water spontaneously. So, if it goes spontaneously probably the nanobubbles just accelerate the mass balance between the solid and solution. The question is what about mixing of vaterite in water without bubbles? Can you really control the absence of nanobubbles?
2) Is air bubbling in crucial for the process? Is it possible to do the process in argon or nitrogen? On the other hand what about pure oxygen?
3) In the conclusion, authors mentioned the size and bubbling rate. Is it possible to do any comparison?
Author Response
请看附件
